# Co-Occurrence Analysis of Citrus Root Bacterial Microbiota under Citrus Greening Disease

**DOI:** 10.3390/plants13010080

**Published:** 2023-12-26

**Authors:** Jong-Won Park, W. Evan Braswell, Madhurababu Kunta

**Affiliations:** 1Citrus Center, Texas A&M University-Kingsville, 312 N. International Blvd., Weslaco, TX 78599, USA; 2Insect Management and Molecular Diagnostic Laboratory, USDA APHIS PPQ S&T, Edinburg, TX 78541, USA

**Keywords:** Huanglongbing, citrus, 16S metagenomics, OTU clustering, microbial co-occurrence network

## Abstract

*Candidatus* Liberibacter asiaticus (CLas) is associated with Citrus Huanglongbing (HLB), a devastating disease in the US. Previously, we conducted a two-year-long monthly HLB survey by quantitative real-time PCR using root DNA fractions prepared from 112 field grapefruit trees grafted on sour orange rootstock. Approximately 10% of the trees remained CLas-free during the entire survey period. This study conducted 16S metagenomics using the time-series root DNA fractions, monthly prepared during twenty-four consecutive months, followed by microbial co-occurrence network analysis to investigate the microbial factors contributing to the CLas-free phenotype of the aforementioned trees. Based on the HLB status and the time when the trees were first diagnosed as CLas-positive during the survey, the samples were divided into four groups, Stage H (healthy), Stage I (early), II (mid), and III (late) samples. The 16S metagenomics data using Silva 16S database v132 revealed that HLB compromised the diversity of rhizosphere microbiota. At the phylum level, Actinobacteria and Proteobacteria were the predominant bacterial phyla, comprising >93% of total bacterial phyla, irrespective of HLB status. In addition, a temporal change in the rhizosphere microbe population was observed during a two-year-long survey, from which we confirmed that some bacterial families differently responded to HLB disease status. The clustering of the bacterial co-occurrence network data revealed the presence of a subnetwork composed of *Streptomycetaceae* and bacterial families with plant growth-promoting activity in Stage H and III samples. These data implicated that the *Streptomycetaceae* subnetwork may act as a functional unit against HLB.

## 1. Introduction

Huanglongbing (HLB; also known as citrus greening) is one of the most destructive diseases in citrus and is caused by three Gram-negative phloem-limited α-Proteobacteria, *Candidatus* Liberibacter asiaticus (CLas), Ca. L. americanus (CLam), and Ca. L. africanus (CLaf) [1,2,3,4]. In the United States, HLB, caused by CLas, was first reported in Florida in 2005 [5] and in Texas and California in 2012 [6,7]. CLas-infected trees develop various symptoms ranging from blotch mottling on the leaf to branch die-back, severely compromising the health of the infected trees [8]. Since the first HLB case was reported in Florida in 2005, the citrus industry in Florida has suffered a significant yield reduction, causing severe economic losses [9]. Interestingly, although there is no known commercial citrus cultivar resistant to HLB, the presence of healthy-looking trees, commonly known as HLB “escape trees”, have been reported in the field together with neighboring HLB-positive trees [10,11]. Considering that both escape trees and HLB-positive trees were found in the same field under the same agricultural practices, including irrigation and pesticide and fertilizer application, it was speculated that one of the factors contributing to the healthy-looking phenotype of those escape trees could be derived from the variation in the microbe composition in the root system [10,11,12]. According to Trivedi et al. [13], the change in the composition of bacterial community in the root system due to CLas infection took place both in quantitative and qualitative ways, which could be the outcome of the interaction among different microbe populations in the root system responding to the disease status. As an attempt to investigate the potential of beneficial rhizosphere microbes as a tool to mitigate the damage caused by CLas infection in citrus, Riera et al. [10] isolated bacteria exhibiting antimicrobial activity from the rhizosphere of the escape trees.

Recently, numerous reports suggested that the root-associated microbiota can influence the fitness of a plant, a sessile organism, in harsh environmental conditions [14,15,16,17,18,19,20,21,22]. The structure of rhizosphere microbiota can be altered by root exudates, which subsequently affects the plant phenotype toward various biotic and abiotic stresses [15,16,17,18,23,24,25]. These findings have opened a new research area for the plant science community to seek a novel way to improve the overall plant performance under various stress conditions by manipulating the composition of root-associated microbiome not only for stress resistance but also for the sustainability of the current agriculture system [17,20,26,27,28,29,30,31]. However, it is not an easy task to achieve, as the structure of rhizosphere microbiota changes in response to environmental factors in the field [16] and more importantly, because the majority of plant metagenomics data available provide only a snapshot of microbial structure, lacking the information to examine the temporal interaction dynamics among microbe population responding to environmental stimuli imposed onto host plants [27].

Previously, we conducted a two-year-long monthly HLB survey on Rio Red grapefruit trees (*Citrus* × *paradisi* Macfad.) grafted on sour orange rootstocks (*Citrus aurantium*) in a commercial orchard in the Lower Rio Grande Valley of Texas to monitor the disease progression in the field by quantitative real-time PCR (qPCR) using leaf and fibrous root samples. The HLB spread in the field took place fast that ~70% of trees subjected to the survey became CLas-positive within a year [32]. However, the monthly conducted qPCR assay confirmed that ~10% of the trees subjected to the survey remained CLas-negative during the entire two-year-long survey period. Since the root DNA fractions used for the survey were prepared from the fibrous roots with firmly attached soil particles, it was anticipated that the root DNA fractions contained root-associated microbial DNAs, which could provide an opportunity to investigate the microbe population closely associated with the citrus root system. Since the survey was conducted for twenty-four consecutive months using the monthly prepared root DNA samples, it was speculated that these time-series root DNA fractions could provide a unique opportunity to investigate the temporal dynamics of bacterial microbiota in the root system of the trees with different HLB status or progression history. The current study conducted 16S metagenomics using the monthly prepared root DNA fractions and co-occurrence analysis to examine how HLB has affected the bacterial community over time that is closely associated with the citrus root system and to investigate if there is any microbial factor that may contribute to the phenotype of the escape trees.

## 2. Results

### 2.1. Sample Selection for 16S rDNA-Based Root Microbiome Analysis

To select root DNA samples for the 16S metagenomics study, the twenty-four survey periods were divided into three stages, Stage 1, 2, and 3, each of which consisted of eight survey periods corresponding to early, mid, and late survey periods (Figure 1). A set of ten trees that were diagnosed as CLas-positive by qPCR, each at early (Stage 1), mid (Stage 2), and late (Stage 3) survey periods, were selected and designated as Stage I, II, and III samples, respectively. In addition, a set of ten trees that remained negative for CLas throughout the entire two-year-long survey period were designated as Stage H (Figure 1). Then, from each survey period, a set of ten root DNA samples, each prepared from Stage I, II, III, and H trees, were selected from a collection of monthly prepared root DNA samples that were used for the HLB survey and divided into two groups of five root DNA samples to prepare two DNA pools per stage per survey period, resulting in a total of 192 root DNA pools for 16S metagenomics sequencing (Figure 1).

### 2.2. 16S Metagenomics Sequencing Output and OTU Clustering

Paired-end sequencing of a total of 192 pooled DNA samples (Figure 1) on an Illumina platform followed by quality trimming of the raw sequencing reads generated a total of 53,259,329 reads (Appendix A). The sample filtering step based on the number of reads eliminated two samples from Stage I and one sample each from Stage II and Stage H due to a low number of reads (Appendix A), resulting in a total of 12,480,649 reads for Stage I, 12,868,958 reads for Stage II, 13,713,035 reads for Stage III, and 12,895,766 for Stage H (Appendix A). Among three major OTU clustering methods [33], the current study adopted the open-reference clustering method, which is a combination of reference-dependent clustering (closed-reference clustering) and *de novo* clustering, to overcome the limitation of closed-reference clustering as well as the demand for time and computing power required for *de novo* clustering. OTU clustering of each stage sample was conducted using Silva 16S database v132 (97% of similarity percentage value) with minimum occurrences of two to remove singletons followed by the removal of OTUs with less than a combined abundance of ten. In addition, for the sequences that were not represented in the reference database at a 97% similarity distance, *de novo* OTUs were generated at 80% taxonomic similarity (Table 1). The OTU clustering grouped the reads in 1576 OTUs for Stage I, 1530 OTUs for Stage II, 1816 OTUs for Stage III, and 1776 OTUs for Stage H after removing mitochondrial and chloroplast sequences (Table 1).

### 2.3. Alpha- and Beta-Diversity of Root Bacterial Microbiota of Trees with Different HLB Disease Status

The taxonomic abundance (richness) and diversity of bacterial microbiota in the root samples of each stage were examined based on two alpha-diversity measurements, Chao 1 and Shannon entropy metrics, respectively (Figure 2). The pairwise Mann–Whitney *p*-values of Chao 1 and Shannon indices between Stage I and II and between Stage III and H were >0.05, which suggested no significant differences in the richness and the diversity of bacterial population between Stage I and II and between Stage III and H (Figure 2). On the other hand, the data revealed that both the richness and diversity of the bacterial population in the rhizosphere of those trees that were diagnosed as CLas-positive at early (Stage I) and mid (Stage II) survey periods were compromised when they was compared to the samples that tested positive at the late survey period (Stage III) and the samples that remained CLas-negative (Stage H) (Figure 2). A PCoA plot of beta-diversity based on the Bray–Curtis matrix indicated a large overlap in the bacterial microbiota composition between stages, although the data showed the presence of a slightly wider variation in the diversity of rhizosphere microbiome composition in Stage III and H compared to Stage I and II (Figure 3).

### 2.4. The Composition of Citrus Root Bacterial Microbiota

To obtain an overall insight into rhizosphere microbiota composition established during 24 consecutive monthly survey periods, the bacterial taxa of citrus root microbiota of the trees with different HLB disease statuses or histories (Stage I, II, III, and H) (Figure 1) were examined at the phylum, family, and genus level. At the phylum level, Actinobacteria and Proteobacteria were the predominant bacterial phyla, comprising >93% of the total bacterial phyla, irrespective of HLB status (Figure 4). Actinobacteria were the most prevalent phyla in all four stage samples, ranging from 58.1% to 59.3%, followed by Proteobacteria (35.3% to 35.9%) (Figure 4). About 90% of Proteobacteria belonged to the class of Alphaproteobacteria, and ~8% and ~2% to Gamma- and Deltaproteobacteria, respectively. Bacteroidetes and Chloroflexi had 1.7~1.9% and 1.3~1.5% of the relative abundance, respectively, followed by Acidobacteria (0.8% to 0.9%), Patescibacteria (0.5% to 0.7%), and Gemmatimonadetes (0.4% to 0.5%) (Figure 4). The remaining phyla with less than 0.2% relative abundance included Fibrobacteres, Firmicutes, and Nitrospirae (Figure 4).

At the family level, *Streptomycetaceae* was the most dominant family, comprising 36% to 38% of bacterial families in the root system of trees in all four stages, and was followed by *Dongiaceae* (7.2% to 7.9%), *Streptosporangiaceae* (6.5% to 6.8%), *Sphingomonadaceae* (5.7% to 6.6%), *Rhizobiaceae* (5.6% to 6.4%), *Xanthobacteraceae* (5.4% to 5.8%), *Pseudonocardiaceae* (4.1% to 5.4%), and *Nocardioidaceae* (4.3% to 5.1%) (Figure 5). The bacterial families with 1% to 3% relative abundance included *Micromonosporaceae* (2.1% to 3.1%), *Beijerinckiaceae* (1.1% to 1.3%), *Burkholderiaceae* (1.0% to 1.2%), and *Chitinophagaceae* (0.9% to 1.0%) (Figure 5).

At the genus level, about thirty-one genera had >0.3% relative abundance, twelve of them belonged to Actinobacteria (0.3% to 38%), sixteen to Proteobacteria (0.4% to 8.1%), two to Bacteroidetes (0.5% to 0.8%), and one to Patescibacteria (0.3% to 0.6%) (Figure 6). Among these genera, *Streptomyces* (35.7% to 38%) was the most dominant genus in the citrus root system (Figure 6), and the remaining genera had less than 9% relative abundance, some of which were *Dongia* (7.2% to 8.1%), *Nonomuraea* (6.5% to 6.8%), and *Rhizorhapis* (4.2% to 5%), followed by less abundant genera (1% to less than 4%), two genera in *Xanthobacteraceae* (ambiguous taxa), two *Nocardioidaceae* (*Kribbella* and *Nocardioides*), three *Rhizobiaceae* (*Ensifer*, *Allorhizobium*–*Neorhizobium*–*Pararhizobium*–*Rhizobium,* and one ambiguous taxon), three *Pseudocardiaceae* (*Labedaea*, *Pseudonocardia,* and *Actinophytocoia*), one *Sphingomonadaceae* (*Sphingomonas*), and one *Beijerinckiaceae* (*Microvirga*) (Figure 6).

### 2.5. The Temporal Dynamics of Citrus Root Microbiota in the Trees with Different HLB Status

Since the rhizosphere bacterial composition described in the previous section was just a snapshot of the bacterial microbiota structure in the citrus root system established for 24 months, it lacked the information to examine how the rhizosphere microbe population responded to the HLB disease status over time. To investigate any potential change in citrus root bacterial microbiota over a period of two years in Stage I, II, III, and H samples with different HLB disease histories (Figure 1), the relative abundance (%) of the top twelve bacterial families that had ≥~1% relative abundance were plotted against twenty-four consecutive monthly survey periods (Appendix A). Although there was a fluctuation in the relative abundance of bacterial families during the twenty-four survey periods, the data showed temporal changes (increase or decrease) in the relative abundance of nine of twelve bacterial families, which varied depending on the HLB disease status (Appendix A). For example, when Stage I and Stage H samples were compared, the changes in the relative abundance of nine bacterial families, *Streptomycetaceae*, *Dongiaceae*, *Rhizobiaceae*, *Xanthobacteraceae*, *Pseudocardiaceae*, *Norcardioidaceae*, *Micromonosporaceae*, *Beijerinckiaceae*, and *Burkholderiaceae*, became clear, and they can be grouped into four different patterns: (1) higher increase rate in Stage H than in Stage I (*Streptomycetaceae* and *Beijerinckiaceae*), (2) higher decrease rate in Stage H than in Stage I (*Dongiaceae* and *Xanthobacteraceae*), (3) an increase in Stage H but a decrease in Stage I (*Rhizobiaceae* and *Burkholderiaceae*), and (4) a decrease in Stage H but an increase in Stage I (*Pseudocardiaceae*, *Nocardioidaceae,* and *Micromonosporaceae*) (Appendix A). When the top ten most abundant genera were analyzed against the 24 survey periods, the genera *Streptomyces*, *Dongia*, *Allorhizobium*–*Neorhizobium*–*Pararhizobium*–*Rhizobium*, *Kribella,* and *Labedaea* were the representative genera of the above-mentioned four different patterns of abundance rate change, respectively (Appendix A). These data indicated that the members of the bacterial microbiota closely associated citrus fibrous root system were differentially affected by HLB.

### 2.6. Co-Occurrence Network Analysis of Root Bacterial Microbiota of Trees with Different HLB Status

The top twenty-six most abundant bacterial families, which had >0.3% relative abundance during twenty-four monthly survey periods, were selected from Stage I, II, III, and H datasets for bacterial co-occurrence network analysis (Appendix A). The selected bacterial families included fourteen Proteobacteria, seven Actinobacteria, two Bacteroidetes, one Patescibacteria, one Gemmatimonadetes, and one Chloroflexi (Appendix A). In addition, the 24 consecutive survey number was included in the co-occurrence network analysis as feature metadata to examine how the time factor affected the bacterial co-occurrence network over time (i.e., 24 survey periods).

The co-occurrence network analysis captured 50 associations among 21 nodes (bacterial families only) in Stage I, 34 associations among 22 nodes (21 bacterial families and the survey number) in Stage II, 70 associations among 23 nodes (22 bacterial families and the survey number) in Stage III, and 85 associations among 24 nodes (23 bacterial families and the survey number) in Stage H (Table 2; Figure 7 and Appendix A).

The co-occurrence network data showed that the twenty-four survey periods had no (Stage I) or limited (Stage II and III) impact on the co-occurrence network of the selected bacterial families (Figure 7A and Appendix A), except for Stage H where the time factor (twenty-four consecutive survey periods) had two positive association (co-occurrence), respectively, with *Streptomycetaceae* (Actinobacteria) and a Patescibacteria (Ambiguous-taxa-21) and five negative association (co-exclusion) with *Dongiaceae* (Proteobacteria), a Chloroflexi (Uncultured-bacterium-20), a Proteobacteria (Uncultured-bacterium-29), *Hyphomicrobiaceae* (Proteobacteria), and *Steroidobacteraceae* (Proteobacteria) (Figure 7B and Appendix A).

The co-occurrence analysis also revealed the hub families that had a high number (degree) of connections with other bacterial families (Figure 7 and Appendix A; Appendix A). The Stage I co-occurrence network showed that seven bacterial families, *Gemmatimonadaceae*, Uncultured-bacterium-29 (Proteobacteria), *Xanthobacteraceae*, *Hyphomicrobiaceae*, *Dongiaceae*, *Steroidobacteraceae,* and *Reyranellaceae*, had six to nine degrees of interactions with other bacterial families (Figure 7A; Appendix A). Among these, the nature of the high degree of interaction among the following five families, *Gemmatimonadaceae*, a Proteobacterium (Uncultured-bacterium-29), *Xanthobacteraceae*, *Hyphomicrobiaceae,* and *Dongiaceae*, were also maintained in the Stage II, III, and H co-occurrence network (Appendix A). However, the number of positive associations between different root bacterial families in the network greatly varied when the Stage I and II network was compared to the Stage III and H network (Figure 7 and Appendix A; Appendix A). Unlike Stage I and II, the co-occurrence network of Stage III and H showed that *Microscillaceae* and Uncultured-bacterium-20 (Chloroflexi) also had a high degree of interactions, mostly positive interaction, with other bacterial families in the network (Figure 7B and Appendix A; Appendix A). In addition, the Stage III and H network data revealed the high degree of association of *Streptomycetaceae* with other root bacterial families (Figure 7B and Appendix A; Appendix A).

### 2.7. Root Bacterial Subnetwork Involving Streptomycetaceae Affected by HLB

The distinct feature of the root bacterial co-occurrence network of Stage III and Stage H, compared to the network of Stage I and II, was the presence of a subnetwork involving *Streptomycetaceae* that had both positive (co-occurrence) and negative (co-exclusion) associations with other root bacterial families (Figure 7, Figure 8, and Appendix A). Among these, *Streptomycetaceae* had two co-occurrence associations with *Beijerinckiaceae* and *Burkholderiaceae* in Stage III (Figure 9A) and four co-occurrence associations in Stage H with *Beijerinckiaceae*, *Burkholderiaceae*, *Sphingomonadaceae,* and a Patescibacteria (Ambiguous-taxa-21) (Figure 9B). In addition, the data in Figure 8 and Figure 9 show that most of the bacterial groups, that had a co-exclusion association with *Streptomycetaceae* in Stage III and H, had a high degree of co-occurrence associations with other root bacterial families in the network, including *Caulobacteraceae*, *Dongiaceae*, *Gemmatimonadaceae*, *Hyphomicrobiaceae*, a Chloroflexi bacteria (Uncultured-bacterium-20), a Proteobacteria (Uncultured-bacterium-29), *Reyranellaceae*, *Solirubrobacteraceae*, and *Steroidobacteraceae* (Figure 8 and Figure 9; Appendix A).

The bacterial co-occurrence network analysis data showed that the degree of associations of *Streptomycetaceae* with other bacterial families decreased from Stage H to Stage I (Figure 7, Figure 9, and Appendix A). Considering that the grouping of trees (Stage I, II, III, and H) was based on the HLB status or when the trees were diagnosed as CLas-positive during the two-year-long HLB survey (Figure 1), the data suggested that the bacterial co-occurrence network involving *Streptomycetaceae* was greatly affected by HLB.

## 3. Discussion

It is now well perceived that the plant-associated microbe population, the outcome of the co-evolution of plant host and its associated microbes, is one of the major integral components contributing to plant adaptation to various biotic and abiotic stress conditions [16,20,21,22,25,34,35]. The soil is considered a major source of microbial population associated with plant organs both above and below ground that encounter continuous fluctuation of environmental stimuli [16,35,36,37]. Plants can selectively recruit rhizosphere microbes with their root exudates derived from photosynthetically fixed carbon sources to facilitate plant growth under various stress conditions [17,20,38]. The microbe population associated with the plant root system exerts a crucial role in maintaining plant health and productivity, which depends on the interaction type that the root microbe community has with the plant [21,39,40,41]. This indicated that the phenotypic characteristics of a crop species that can be observed in the field are influenced by not only the plant genotype but also the microbes associated with the plant root system [21,22,39].

Even though there were no commercial citrus cultivars resistant to HLB available to growers thus far, the occurrence of trees with healthy-looking phenotypes has been reported among trees severely affected by HLB in the field [10,11]. We also observed a similar phenomenon from our previous HLB survey conducted monthly for two years in a commercial grapefruit orchard in Texas. In this survey, we confirmed that ~10% of the trees subjected to the survey remained disease-free during the entire two-year-long survey conducted by qPCR using DNA fractions prepared from leaves and fibrous roots. It has been shown that HLB compromised the recruitment and/or the composition of certain microbes in the citrus root system [12,42], and an attempt has been made to use bacteria with antibacterial activity isolated from the rhizosphere of the escape trees as a tool to mitigate the damage caused by HLB [10]. These data prompted us to investigate and compare the microbe population associated with the fibrous root system of trees with different HLB disease histories to investigate the potential involvement of rhizosphere microbe(s) for the tolerance against HLB as seen in the escape trees. Since the root DNA fractions prepared monthly during the two-year-long HLB survey were prepared from fibrous root samples with firmly attached soil particles, it was anticipated that these root DNA samples should contain the microbial DNAs of microbes closely associated with fibrous root samples. In addition, since the root DNA fractions were prepared monthly from the same trees for twenty-four consecutive months, the current study was able to investigate the temporal changes of the rhizosphere bacterial population of trees with different HLB disease histories unlike most other rhizosphere metagenomics studies that provided a snapshot of information on the rhizosphere microbiome structure [27].

The 16S metagenomics data obtained from the aforementioned time-series root DNA samples with different HLB disease histories confirmed that the diversity of the rhizosphere bacterial population was compromised by HLB. They also revealed that some root-associated bacterial families reacted differently to HLB. For example, as time passes, the relative abundance of *Streptomycetaceae* and *Beijerinckiaceae* increased faster in the root system of CLas-negative trees (Stage H) compared to the trees diagnosed as HLB at early stage of the survey (Stage I), while an increase in the relative abundance of *Rhizobiaceae* and *Burkholderiaceae* was observed only in the root system of CLas-negative trees. These findings were intriguing as these four bacterial families are known to include bacteria with antibacterial and plant growth-promotion activity [43,44,45,46].

The microbial co-occurrence network analysis revealed that the subnetwork formed with *Streptomycetaceae* in Stage H and III samples had a positive correlation with time factor (twenty-four consecutive survey periods), indicating that the relative abundance of *Streptomycetaceae* increased over time, unlike the samples in Stage I and II. In addition, the direct co-occurrence association of *Streptomycetceae* with *Beijerinckiaceae* and *Burkholderiaceae* and indirectly with *Rhizobiaceae* in the samples of Stage H and III suggested that the microbes included in the co-occurrence subnetwork of *Streptomycetaceae* may play a role as a functional unit related to the phenotypes of samples in Stage H (CLas-negative) and samples in Stage III (delayed CLas infection). Interestingly, *Streptomycetaceae* in Stage H and Stage III networks had co-exclusion association with major bacterial hub families that had co-presence association with many other rhizosphere bacterial families such as *Caulobacteraceae*, *Dongiaceae*, *Gemmatimonadaceae*, *Hyphomicrobiaceae*, *Reyranellaceae,* and *Solirubrobacteraceae*. According to these data and the data obtained with samples diagnosed as HLB-positive at early (Stage I) and mid (Stage II) survey periods, which lacked the *Streptomycetaceae* co-occurrence subnetwork, HLB greatly compromised the bacterial co-occurrence subnetwork involving *Streptomycetaceae* in the diseased trees. This study suggested a potential role of microbes included in the *Streptomycetaceae* subnetwork in maintaining CLas-negative status, as seen in Stage H samples.

To our knowledge, this is the first citrus metagenomics study conducted with time-series samples (i.e., monthly prepared root DNA fractions) that has provided a unique opportunity to examine the temporal dynamics of bacterial interaction network associated in the citrus root system through microbial co-occurrence network analysis. Although the current study does not provide direct evidence for the involvement of the bacterial interaction subnetwork involving *Streptomycetaceae* in maintaining the healthy phenotype of the escape trees, the data obtained in this study do not rule out this possibility. Further study to examine the functional relevancy of the subnetwork consisting of *Streptomycetaceae* will provide better insight into the functionality of the microbes included in the subnetwork in maintaining plant health and productivity under the current HLB endemic situation in the field.

## 4. Materials and Methods

### 4.1. Root DNA Samples

The root DNA samples used in this study were derived from our previous two-year-long monthly HLB survey that was conducted in a ca. 2-hectare block of 4–5 years old grapefruit trees on sour orange rootstock in a commercial orchard in Texas (US plant hardiness zone 10a) [32]. Fibrous root samples were collected from two locations of each tree, about two feet from the tree trunk, by digging one to five inches deep into the soil (sandy clay loam) [32]. Prior to root DNA extraction, the collected root samples were air dried for 24 h at room temperature from which excess big soil particles were removed from the fibrous roots by gentle tapping with fingertips. The root samples were chopped to 1–2 mm in length to aid maceration followed by root DNA extraction following the manual of the DNeasy PowerPlant Pro HTP96 kit (Qiagen, Hilden, Germany) [32]. The prepared root DNA fractions were kept in a −20 °C freezer.

### 4.2. 16S Metagenomics Data Analysis

The root DNA fractions prepared monthly from a set of 10 trees of Stage I, Stage II, Stage III, and Stage H were divided into 2 groups, each of which was composed of 5 root DNA samples, to prepare 2 root DNA pools of each stage sample at each survey period (Figure 1). A total of 400 ng of each root DNA sample was used for the DNA pool. A total of 192 DNA pools (2 root DNA pools × 4 stages × 24 survey periods) were used for 16S metagenomics sequencing at Genewiz using its 16S-EZ sequencing platform that targets the V3 and V4 16S rRNA genes.

The 16S metagenomics data analysis was conducted following the amplicon-based analysis workflow in the CLC Microbial Genomics Module (Qiagen, Aarhus, Denmark). Briefly, the paired-end raw sequencing reads were quality-filtered using a tool with a default setting in the CLC Microbial Genomics Module (Qiagen, Aarhus, Denmark) from which the samples with a low number of reads were filtered out by a filtering tool based on the number of reads in the CLC Microbial Genomics Module. Then, OTU clustering of the filtered samples was conducted using Silva 16S database v132 (97% of similarity percentage value) with minimum occurrences of 2 to remove singletons followed by OTU removal with less than the combined abundance of 10. In addition, *de novo* OTUs were generated at 80% taxonomic similarity using sequences not mapped to the reference database. Then, the low-abundance OTUs were eliminated from the OTU table following a default setting from which a taxonomic profiling abundance table was generated. The alpha and beta diversities were estimated in the microbial genomics module after conducting OTU alignment using MUSCLE followed by phylogenetic tree construction.

### 4.3. Microbial Co-Occurrence Network Analysis

The microbial co-occurrence network analysis was conducted using the Co-occurrence Network Inference (CoNet) plug-in in CytoScape v3.10.1 [47,48], from which only OTUs with a minimum occurrence of greater than or equal to 20 were used as input data for CoNet. Pairwise scores were calculated for five methods, Pearson and Spearman correlations, mutual information similarity, and Bray–Curtis and Kullback–Leibler dissimilarity in the Methods menu in CoNet. Permutation set to 1000 was conducted in CoNet by selecting “edgeScores” as routine and “shuffle-rows” as a resampling strategy together with the enabled “Renormalize” option. Then, bootstrapping with 1000 iterations was conducted as the resampling method with a “brown” setting as a merge strategy of *p*-values of the five methods that were computed during the permutation step. In addition, to remove unstable edges outside the 0.95 range of their bootstrap distribution, the option “Filter unstable edges” was enabled. Then, multiple test correction was conducted by enabling the “benjaminihochberg” option in the bootstrap tool in CoNet, from which only edges with FDR-corrected *p*-values of <0.05 were retained. The network was visualized using tools in CytoScape following the CoNet plug-in manual.

## 5. Conclusions

The current study confirmed that HLB compromised both the diversity and richness of bacterial populations closely associated with the citrus fibrous root system. According to the 16S metagenomics data obtained from monthly prepared root DNA fractions for a two-year-long HLB survey, some root-associated bacterial populations differently reacted to HLB over time. The microbial co-occurrence network analysis revealed that the bacterial association subnetwork involving *Streptomycetaceae* and other bacterial families with plant growth-promoting activity was compromised by HLB, suggesting a possibility that the bacterial association involving *Streptomycetaceae* could play a major role in maintaining plant health against HLB. A further follow-up study is needed to examine this possibility.

## Figures and Tables

**Figure 1 plants-13-00080-f001:**
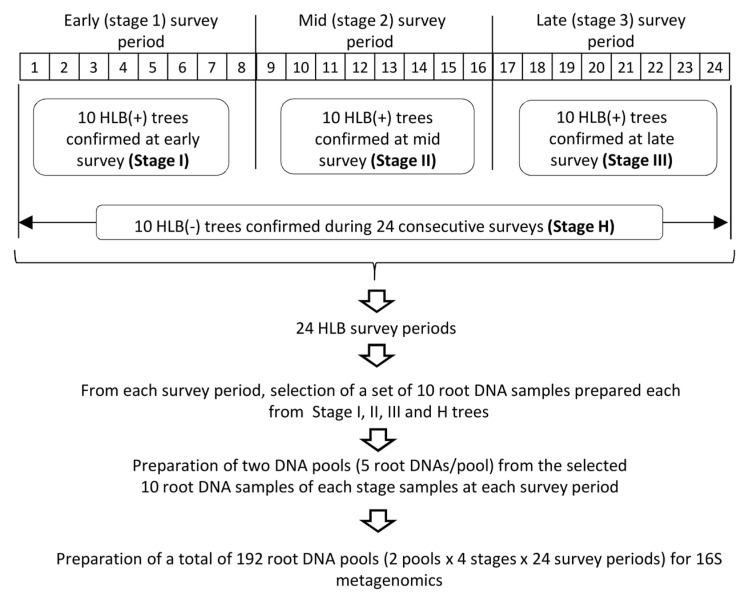
Selection of fibrous root DNA samples for the 16S metagenomics study based on HLB status and the survey period. Twenty-four survey periods were divided into three stages, Stage 1, 2, and 3, each of which consisted of eight survey periods. The trees that tested positive for CLas at the early, mid, and late survey period were designated as Stage I, II, and III. The trees that remained CLas-negative during the entire 2-year-long survey were designated as Stage H.

**Figure 2 plants-13-00080-f002:**
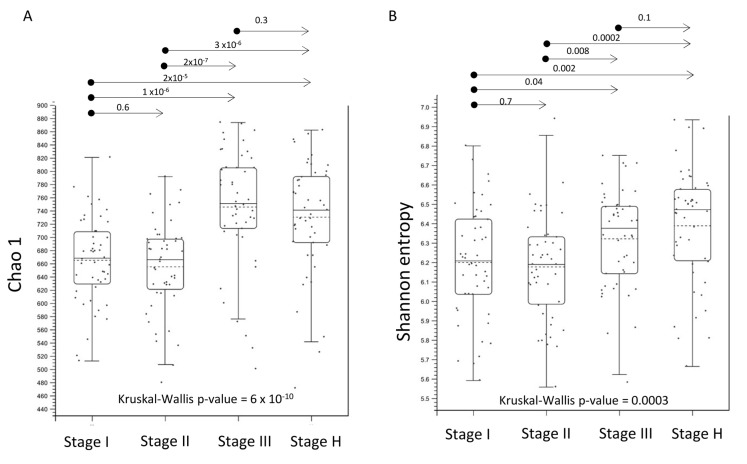
Comparison of alpha-diversity metrics based on Chao 1 (**A**) and Shannon entropy (**B**) indices that measure, respectively, richness and diversity of bacterial microbiota associated with the fibrous root system of citrus trees of Stage I, II, III, and H, each representing samples that tested positive for CLas at early (stage 1), mid (stage 2), and late (stage 3) survey periods and CLas-negative trees (Stage H) during the 2-year-long monthly HLB survey. Kruskal-Wallis *p*-values were indicated in the figure. The *p*-values of a pairwise Mann–Whitney U test were indicated above the arrows in the figure. The median (solid line) and mean (dashed line) in the box plot were indicated.

**Figure 3 plants-13-00080-f003:**
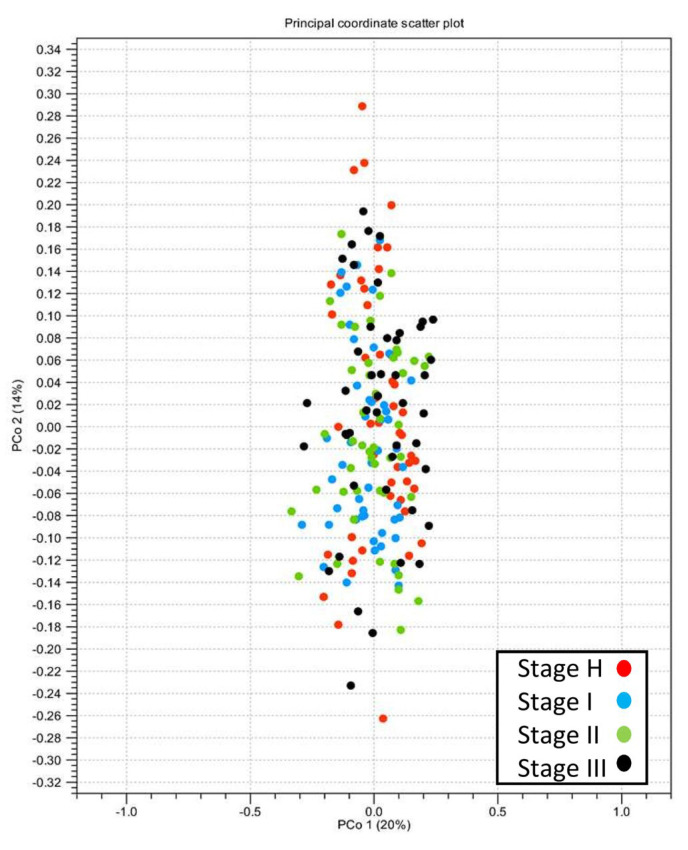
Principal coordinate analysis plot of the beta-diversity of samples of Stage I, II, III, and H based on Bray-Curtis distances.

**Figure 4 plants-13-00080-f004:**
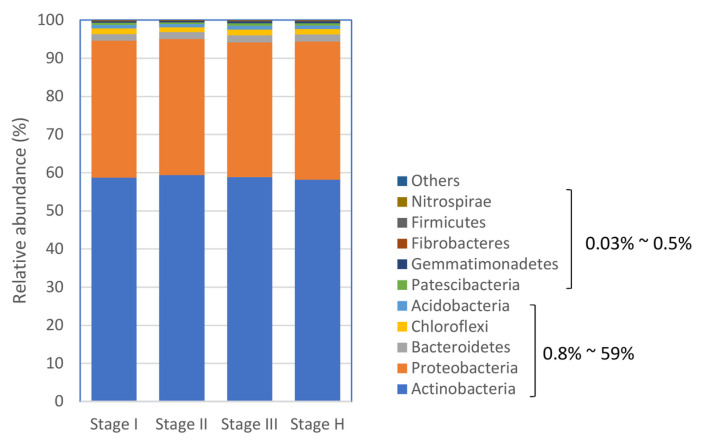
Relative abundance of bacterial phyla in citrus root microbiota based on 16S metagenomics analysis. The range of relative abundance (%) of color-coded phyla was indicated in the figure.

**Figure 5 plants-13-00080-f005:**
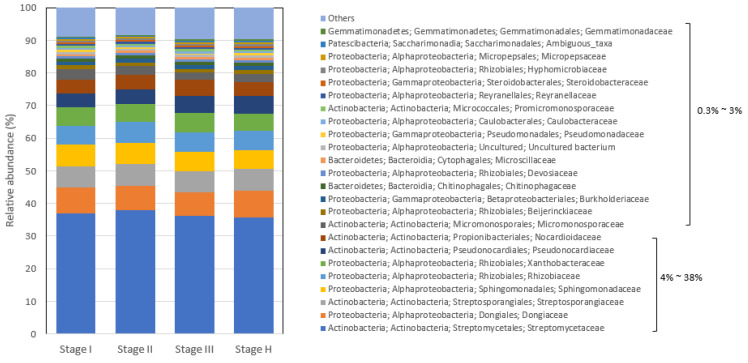
Relative abundance of bacterial families in citrus root microbiota based on 16S metagenomics analysis. The range of relative abundance percentages was indicated in the figure.

**Figure 6 plants-13-00080-f006:**
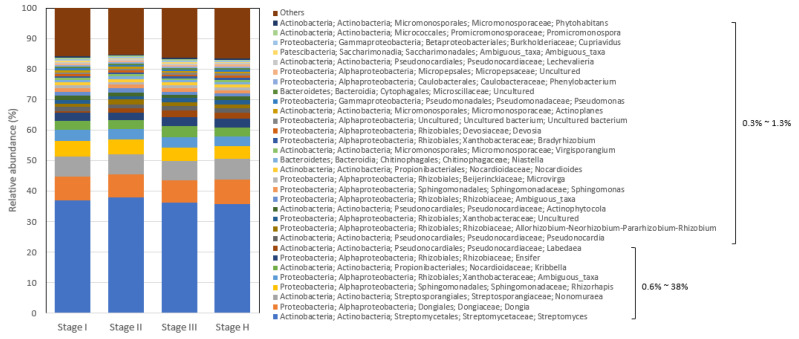
Relative abundance of bacterial genera in citrus root microbiota based on 16S metagenomics analysis. The range of relative abundance percentages was indicated in the figure.

**Figure 7 plants-13-00080-f007:**
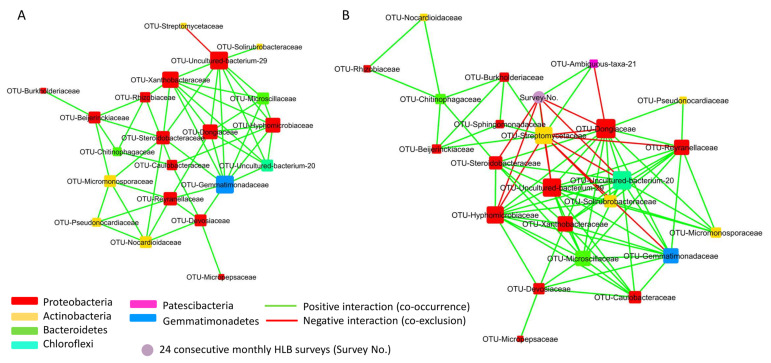
Co-occurrence network analysis of citrus root bacterial microbiota with >0.3% relative abundance at the family level in the trees of Stage I (**A**) and Stage H (**B**). Node size and color indicated the degree (number of connections) and bacterial family, respectively. The positive and negative interactions were indicated in green and red lines, respectively.

**Figure 8 plants-13-00080-f008:**
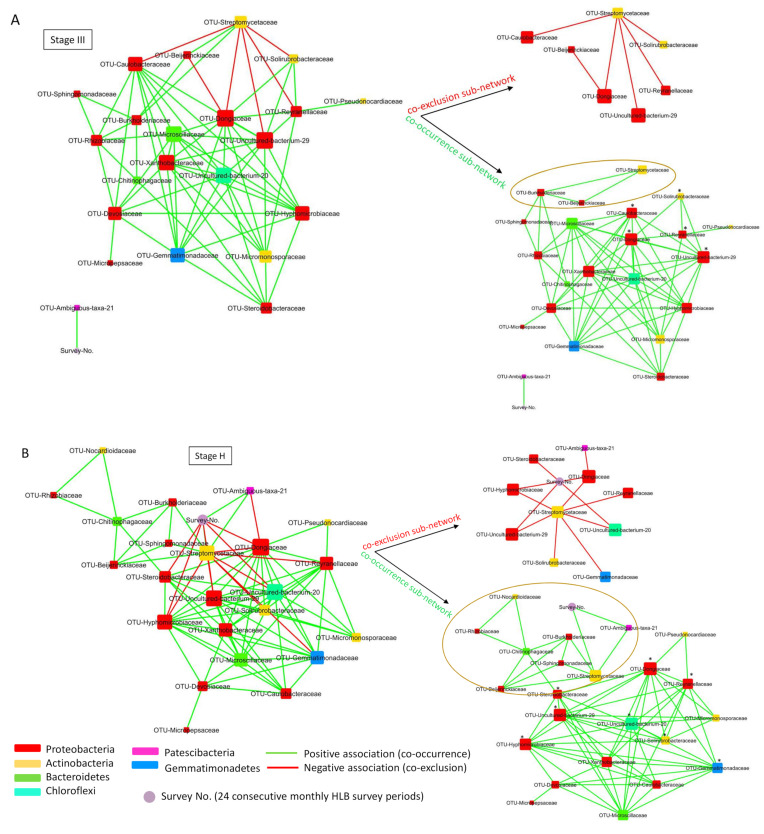
The positive and negative association network involving *Streptomycetaceae* in the co-occurrence network of Stage III (**A**) and Stage H (**B**). Node size and color indicated the degree (number of connections) and bacterial family, respectively. The positive and negative interactions were indicated in green and red lines, respectively. The subnetwork that had a positive association (co-occurrence) with *Streptomycetaceae* was encircled in the figure. * indicated the families that had a negative association with *Streptomycetaceae*, as shown in the co-exclusion subnetwork.

**Figure 9 plants-13-00080-f009:**
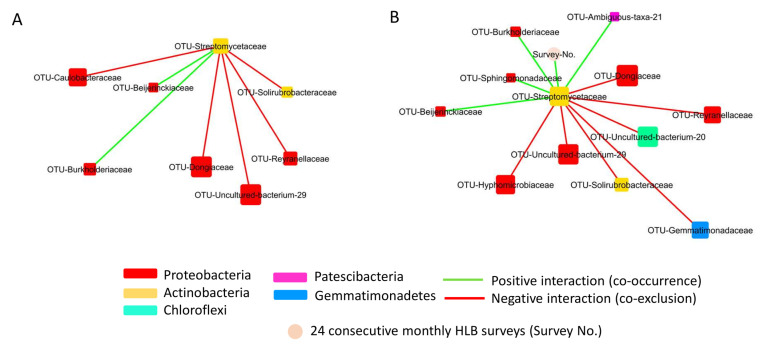
The co-occurrence subnetwork of *Streptomycetaceae* that interacted with other bacterial families resided in the citrus root system. (**A**) The subnetwork of bacterial co-occurrence network of Stage III. (**B**) The subnetwork of bacterial co-occurrence network of Stage H. Node size and color indicated the degree (number of connections) and bacterial family, respectively. The positive and negative interactions were indicated in green and red lines.

**Table 1 plants-13-00080-t001:** Summary of OTU clustering.

Sample ID	Reference Database Size (Silva 16S v132)	No. of OTUs Based on the Database	No. of *de novo* OTUs	Total No. of Predicted OTUs	No. of OTUs after Removing Low Abundance OTUs	No. of OTUs without Mitochondria and Chloroplast Sequences
Stage I	17,222	3655	1496	5151	1598	1576
Stage II		3674	1430	5104	1562	1530
Stage III		3897	2001	5898	1849	1816
Stage H		3808	1975	5765	1809	1776

**Table 2 plants-13-00080-t002:** Summary of root bacterial co-occurrence network analysis.

Edge				Node			
Stage	Total Number of Interactions (Edges)	No. of Positive Interaction (Co-Occurrence)	No. of Negative Interaction (Co-Exclusion)	Stage	Total Number of Nodes *	No. of Nodes with Positive Interaction (Positive Degree (Edge))	Number of Nodes with Negative Interaction (Negative Degree (Edge))
Stage I	50	49	1	Stage I	21	20	2
Stage II	34	34	-	Stage II	22	22	-
Stage III	70	64	6	Stage III	23	23	7
Stage H	85	72	13	Stage H	24	24	11

* A node for the survey number was included in the total number of nodes for Stage II, III, and H.

## Data Availability

Data are contained within the article and Appendix A.

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
