# Peer review of "Co-Occurrence Analysis of Citrus Root Bacterial Microbiota under Citrus Greening Disease"

_plants, 2023, doi:10.3390/plants13010080_

Round 1

Reviewer 1 Report

Comments and Suggestions for Authors

Dear Authors,

I think it's an interesting study. I think this manuscript could be published in Plants.

Kind regards

Author Response

We deeply appreciated Reviewer #1’s recommendation for publication of the manuscript in Plants.

Reviewer 2 Report

Comments and Suggestions for Authors

Comments to the Author:

Upon thorough review of the manuscript concerning the impact of Huanglongbing (HLB) on the diversity of rhizosphere microbial communities in citrus trees, I believe this paper offers valuable insights and makes a significant contribution to this critical field of research. The study utilizes 16S metagenomics data analysis and microbial co-occurrence network analysis to uncover correlations between HLB and specific microbial communities, which may play a pivotal role in plant health. Overall, the theme and experimental design of the paper meet the standards. However, there are areas in the manuscript where data transparency and clarity in the analysis explanations could be improved. Below are my primary concerns as a reviewer and directions for improvement:

1.        The introduction could benefit from the inclusion of a more detailed theoretical foundation on how microbial community dynamics may affect the resistance of citrus trees to HLB. This would aid in a better understanding of why studying the microbial communities within citrus root systems is important for improving HLB management strategies.

2.        It is recommended to incorporate updates on research concerning the effects of microbial community dynamics on plant growth, thereby demonstrating the viability and relevance of this study.

3.        It is advisable to provide a more elaborate explanation of why this study is necessary and the potential significance of such monitoring for understanding and managing HLB.

4.        Line 63-64: “However, ~10% of the trees remained CLas-negative during the entire two year-long survey period (unpublished data)”. Previous research work and unpublished data have been mentioned. For the mention of these data, it is desirable to be able to provide further description or publish these data to support the background information of the current research.

5.        Line 82-84: “were selected and designated as Stage I, II and III samples ... CLas throughout the entire two year-long survey periods were designated as Stage H”. How were these samples selected? How was it determined which samples were HLB-negative and HLB-positive throughout the entire experimental process, especially in distinguishing between Stages I, II and III among the HLB-positive samples?

6.        Line 90: Figure 1 is not clear enough, it is recommended to ensure that all figures and tables are clear when submitting the final manuscript so that readers can easily follow and understand the description of the manuscript.

7.        2.2.16. S Metagenomics Sequencing Output and OTU Clustering: The obtained data and OTU clustering method are described. It is recommended to provide a short note discussing the choice of OTU clustering method and its advantages.

8.        2.3. Alpha- and Beta-Diversity of Root Bacterial Microbiota of Trees with Different HLB Disease Status: It is recommended to provide specific data of alpha-diversity, rather than just mention high and low, which will help to understand the data changes more intuitively.

9.        2.4. The Composition of Citrus Root Bacterial Microbiota: Although the abundance of each major bacterial phylum is mentioned, the specific effects of these bacteria on the health of citrus trees are not discussed, especially under different HLB conditions. Consideration could be given to increasing discussion of how to interpret these data and their possible ecological or pathological implications.

10.    Starting with line 141, the original 2.4 was incorrectly labeled as 2.3, and the subsequent numbering needs to be adjusted accordingly.

11.    The analysis of data in the whole manuscript is very weak, especially the lack of combined analysis with existing relevant studies and systematic analysis of the phenomena found in the data.

Comments on the Quality of English Language

The manuscript is fluent in English and easy to read

Author Response

Thank you for the constructive comments. Below I have copied those comments in bold italics, with our response underneath. As can be seen below, essentially all issues have been dealt with by author response and by incorporating changes or additions directly in the text of the manuscript.

Reviewer 3 Report

Comments and Suggestions for Authors

Comments and suggestions for authors:

Comments and suggestions for authors:

This research supplied some useful and novel scientific information having valuable references. The study showed a deep background, careful conducting, and good data.  Some issues below should be considered for addressing:

-        The abstract is too long (384 words), it should be reduced to approximately 250 words.

-        The new findings and significant results of this work should be clearly indicated.

-        How to identify Huanlongbing (HLB) containing in seek citrus?

- Figure 1, figure 8, and figure 9 should be provided at higher quality to better observation.

Comments on the Quality of English Language

The Enligh is be checked.

Author Response

(The authors gave the same response as above.)
